# BREAKING THE MEMORY BARRIER: NEAR INFINITE BATCH SIZE SCALING FOR CONTRASTIVE LOSS

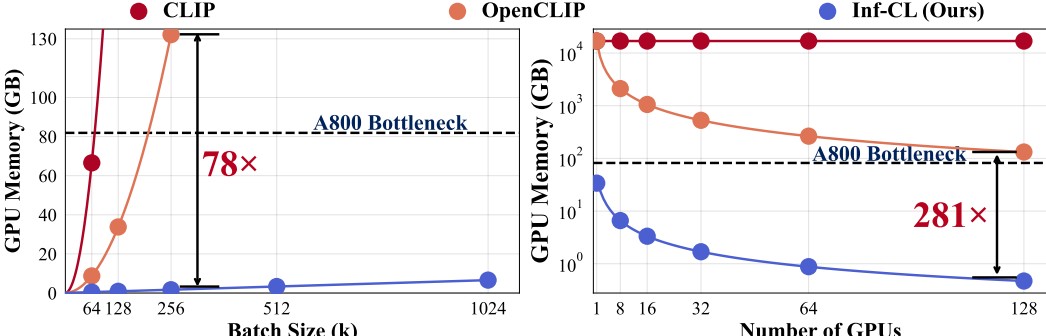

Figure 1: **GPU memory usage comparison** between **Inf-CL** and previous methods (**CLIP**, **Open-CLIP**). The dashed line marks the common GPU memory limit. Memory costs exceeding the bottleneck of 80G A800 are estimated by curve fitting. ❶ Left: With 8×A800, CLIP and OpenCLIP's memory consumption increases quadratically, while Inf-CL achieves linear growth, reducing memory costs by **78×** at a batch size of 256k. ❷ Right: At a batch size of 1024k, even with 128 GPUs, previous methods exceed memory limits, whereas Inf-CL reduces memory demand by **281×**.

## ABSTRACT

Contrastive loss is a powerful approach for representation learning, where larger batch sizes enhance performance by providing more negative samples to better distinguish between similar and dissimilar data. However, scaling batch sizes is constrained by the quadratic growth in GPU memory consumption, primarily due to the full instantiation of the similarity matrix. To address this, we propose a tile-based computation strategy that partitions the contrastive loss calculation to arbitrary small blocks, avoiding full materialization of the similarity matrix. Furthermore, we introduce a multi-level tiling strategy to leverage the hierarchical structure of distributed systems, employing ring-based communication at the GPU level to optimize synchronization and fused kernels at the CUDA core level to reduce I/O overhead. Experimental results show that the proposed method scales batch sizes to unprecedented levels. For instance, it enables contrastive training of a CLIP-ViT-L/14 model with a batch size of 4M or 12M using 8 or 32 A800 80GB without sacrificing any accuracy. Compared to SOTA memory-efficient solutions, it achieves a two-order-of-magnitude reduction in memory while maintaining comparable speed. The code will be made publicly available.

## 1 INTRODUCTION

Contrastive learning serves as a foundational technique across various applications, such as multi-modality retrieval (Radford et al., 2021; Luo et al., 2022; Girdhar et al., 2023), self-supervised representation learning (Chen et al., 2020a; He et al., 2020; Gao et al., 2022), and dense text retrieval (Wang et al., 2022). It learns an embedding space in which similar data pairs stay close while dissimilar ones are far apart (Hadsell et al., 2006; Oord et al., 2018; Weng, 2021). Large batch sizes are critical to the success of contrastive learning due to their reliance on in-batch negatives (Chen

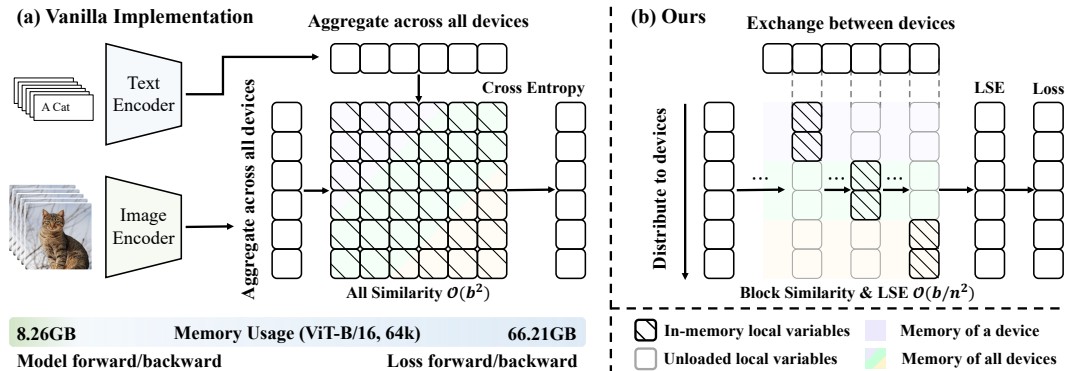

Figure 2: (a) **Vanilla implementation of contrastive loss** gathers features to all devices to calculate all similarity simultaneously, where the similarity with squared complexity are repeatedly stored in all devices, causing huge memory costs for loss calculation when batch size increases. (b) **Our Inf-CL** significant decreases the memory cost by serial and distributed tile-wise computation.

et al., 2020a; Radford et al., 2021). Specifically, larger batches provide a diverse set of negative samples, enhancing the model's ability to learn discriminative representations (Pham et al., 2021).

Despite the above benefits, scaling batch size in contrastive learning is severely limited by GPU memory. The memory required for computing and storing image-text similarity matrices (see Figure 2(a)) grows quadratically with batch size, making further scaling impractical and limiting the potential performance gains, even with advanced hardware. Several methods have been proposed to address memory limitations in scaling batch sizes for contrastive learning. Gradient-Cache (Gao et al., 2021) reduces memory usage by decoupling model and loss computations, but the memory cost of the loss remains a significant bottleneck. OpenCLIP (Ilharco et al., 2021) and DisCo-CLIP (Chen et al., 2023) improve efficiency by distributing contrastive loss computation across $n$ GPUs, reducing memory consumption by a factor of $n$. However, despite advances in memory-efficient techniques, most studies are limited to a batch size of $128k$, restricting the potential of contrastive learning and the scaling demands of modern models and datasets (Saunshi et al., 2019; Chen et al., 2022; Kaplan et al., 2020).

In this paper, we introduce **Inf-CL**, a novel approach to mitigate the quadratic memory cost in contrastive learning, which is caused by the full instantiation of the similarity matrix for log-sum-exp (LSE) computation. Instead of storing the entire matrix, **Inf-CL** partitions the LSE calculation into smaller, sequentially computed tiles, leveraging the cumulative property of LSE. This confines memory usage to the tile size and the number of parallel tiles, allowing for a trade-off between memory and computational efficiency. To enhance practical efficiency, we propose a multi-level tiling strategy. At a coarse-grained level, image and text batches are distributed across multiple GPUs, with each GPU performing serial LSE computations on multiple rows. As computations proceed, asynchronous column-wise data exchange minimizes communication overhead, as illustrated in Figure 2(b). At a fine-grained level, row-wise computations are parallelized across CUDA cores within each GPU, consolidating iterations into a single kernel to reduce I/O overhead. Theoretically, **Inf-CL** can compute contrastive loss with nearly infinite batch sizes using a small tile size, albeit with reduced speed. The multi-level tiling strategy is crucial to achieving practical scalability and efficiency, balancing memory reduction with computation speed.

We evaluate **Inf-CL** on the image-text contrastive learning task. As shown in Figure 1, **Inf-CL** reduces space complexity from quadratic (e.g., $\mathcal{O}(b^2)$ for CLIP, $\mathcal{O}(b^2/n)$ for OpenCLIP) to linear ($\mathcal{O}(b/n^2)$ for **Inf-CL**), where $b$ is the batch size and $n$ is the number of GPUs. This significant reduction in memory cost enables efficient training of large batch sizes. For instance, training a ViT-L/14 CLIP model with a batch size exceeding 10M on 32 A800 GPUs (80 GB each) requires only 1.44 GB of memory per GPU—over a **30×** improvement compared to previous methods. Moreover, **Inf-CL** maintains precision consistent with existing approaches, ensuring no loss in accuracy. In terms of computation time, **Inf-CL** matches the performance of prior methods, requiring approximately 59 hours to process a 64k batch size on 8 A800 GPUs. The time cost scales nearly linearly with batch size, as demonstrated by a batch size increase from 64k to 256k resulting in a roughly 4× growth in training time ($220.3/49.4 \approx 4$).

In summary, our contributions include:

- We propose a tile-based contrastive loss implementation that iteratively accumulates the LSE term, removing the need to instantiate the full similarity matrix and significantly reducing memory overhead. This approach theoretically allows training with nearly infinite batch sizes using sufficiently small tiles.

- We propose a multi-level tiling strategy for a distributed training system, which reasonably leverages parallelism to achieve a balance between memory and computational efficiency.

- Our experiments demonstrate that **Inf-CL** scales batch sizes to unprecedented levels (e.g., 12M for CLIP-ViT-L/14 on 32 A800 80GB GPUs) while maintaining accuracy and comparable training speed to state-of-the-art methods.

## 2 PRELIMINARIES

### 2.1 DISTRIBUTED TRAINING SYSTEM

**Cross-GPU Communication:** For scaling batch size, training across multiple GPUs is crucial to handle memory and computational demands. However, communication overhead between GPUs can limit performance. Techniques like hierarchical all-reduce and ring-based communication alleviate such overhead by optimizing synchronization between GPUs (Liu et al., 2023). Blockwise parallelism, as employed in methods like ring attention, further improves efficiency by overlapping computation and communication.

**GPU Memory and Execution:** The performance of modern deep learning models relies heavily on hardware resources, particularly GPU memory and execution capabilities. GPUs, like A100s, typically have two different types of memory: HBM (High Bandwidth Memory) and SRAM (Static Random Access Memory). HBM serves as the primary memory with a capacity of up to 80GB. In contrast, SRAM is much smaller (usually measured in megabytes) but offers a significantly faster access speed, acting as a vital cache for frequently accessed data and enabling rapid computations. Techniques like FlashAttention (Dao et al., 2022) show that fine-grained control over the memory access of HBM and the fuse the operations can achieve faster training and less memory usage.

### 2.2 VANILLA IMPLEMENTATION OF CONTRASTIVE LOSS

In contrastive learning, the objective is to learn an embedding space where similar samples (positive pairs) are pulled closer, while dissimilar samples (negative pairs) are pushed away. A typical implementation, exemplified by CLIP (Radford et al., 2021), is depicted in Figure 2. The image and text encoders are trained with contrastive loss after extracting features. For brevity, we only discuss image-to-text contrastive loss as an example in the following sections, since the implementation of text-to-image contrastive loss is symmetric. Specifically, given a batch size of $b$ and the in-batch $c$-dimensional visual feature $\boldsymbol{I} \in \mathbb{R}^{b \times c}$ and textual feature $\boldsymbol{T} \in \mathbb{R}^{b \times c}$, the image-to-text contrastive loss is defined as

$$\mathcal{L}_I = -\frac{1}{b} \sum_{i=1}^{b} \log \frac{e^{x_{i,i}}}{\sum_{j=1}^{b} e^{x_{i,j}}} \tag{1}$$

where $x_{i,j} = \boldsymbol{I}_i \cdot \boldsymbol{T}_j$ is the scaled cosine similarity between the $i$-th image and $j$-th text, and $x_{i,i}$ represents the positive pair. Here, we omitted the temperature factor for simplicity.

The vanilla implementation first computes the similarity matrix $\boldsymbol{X} \in \mathbb{R}^{b \times b} = \boldsymbol{I} \cdot \boldsymbol{T}'$ and stores it in high-bandwidth memory (HBM). Afterward, softmax normalization followed by the calculation of negative log-likelihood is applied to the similarity matrix. The memory required to store $\boldsymbol{X}$ and its normalized results scales as $\mathcal{O}(b^2)$, which can occupy a substantial amount of GPU memory when $b$ is large. Figure 2 gives an example of training ViT-B/16 with a batch size of 64k, using model memory optimization techniques such as Gradient Cache (Gao et al., 2021; Pham et al., 2021). As can be seen, the GPU memory footprint of the model itself is only 5.24GB while the loss calculation still requires 66GB. This indicates that, with batch size scaling, the memory bottleneck during training lies in the loss calculation. Although large batch sizes are necessary for improving model performance (Saunshi et al., 2019; Chen et al., 2022), the traditional implementation struggles to support them due to excessive memory consumption in the loss calculation.

## 3 METHOD

### 3.1 TILE-WISE CONTRASTIVE LEARNING

As discussed in Section 2.2, the root cause of the quadratic memory growth in the vanilla implementation is the full materialization of the similarity matrix $\boldsymbol{X}$. To eliminate the memory cost, we first decompose the operations related to $\boldsymbol{X}$ from the loss function:

$$\mathcal{L}_I = -\frac{1}{b}\sum_{i=1}^{b}(x_{i,i} - \log\sum_{j=1}^{b}e^{x_{i,j}}) = -\frac{1}{b}\sum_{i=1}^{b}x_{i,i} + \frac{1}{b}\sum_{i=1}^{b}\log\sum_{j=1}^{b}e^{x_{i,j}}, \tag{2}$$

where the spatial complexity of the first part is $\mathcal{O}(b)$, and for the second log-sum-exp (LSE) part, it is $\mathcal{O}(b^2)$. Based on this formulation, we introduce a tile-wise contrastive learning method that avoids the full materialization of $\boldsymbol{X}$ by iterative accumulation between tiles. The following sections provide a detailed formulation of the forward and backward processes.

**Tile-Wise Forward.** To reduce the dependency on storing $\boldsymbol{X}$ entirely, we adopt a tile-wise approach for calculating $\boldsymbol{l}$. The process is show as below:

$$\underbrace{\begin{bmatrix} \boldsymbol{X}^{1,1} & \cdots & \boldsymbol{X}^{1,n_c} \\ \vdots & \ddots & \vdots \\ \boldsymbol{X}^{n_r,1} & \cdots & \boldsymbol{X}^{n_r,n_c} \end{bmatrix}}_{\text{Tiled computation of } \boldsymbol{X}} \rightarrow \underbrace{\begin{bmatrix} \boldsymbol{l}^{1,1} & \cdots & \boldsymbol{l}^{1,n_c} \\ \vdots & \ddots & \vdots \\ \boldsymbol{l}^{n_r,1} & \cdots & \boldsymbol{l}^{n_r,n_c} \end{bmatrix}}_{\text{Merged serially via Eq. 4}} \rightarrow \begin{pmatrix} \boldsymbol{l}^1 \\ \vdots \\ \boldsymbol{l}^{n_r} \end{pmatrix} = \boldsymbol{l} \tag{3}$$

where $n_r$ and $n_c$ represent the number of tiles along the rows and columns, respectively. The computation proceeds by dividing $\boldsymbol{X}$ into multiple tiles, denoted as $\boldsymbol{X}^{i,j}$, and then calculating the intermediate LSE values $\boldsymbol{l}^{i,j} = \text{LSE}(\boldsymbol{X}^{i,j})$ within each tile. The resulting LSE values from each column of tiles are then merged serially along the rows to obtain the final global LSE vector $\boldsymbol{l}$.

To prevent numerical instability and overflow during the merging process, the following numerically stable operation is performed:

$$\boldsymbol{l}^i \leftarrow \boldsymbol{l}^i + \log(1 + e^{\boldsymbol{l}^{i,j} - \boldsymbol{l}^i}), \ j = 1, \ldots, n_c, \tag{4}$$

where the initial value of $\boldsymbol{l}^i$ is 0. In each iteration, the intermediate value $\boldsymbol{l}^{i,j}$ is merged with $\boldsymbol{l}^i$, and after processing all $n_c$ tiles, the global LSE vector $\boldsymbol{l}$ is obtained.

During the computation of $\text{LSE}(\boldsymbol{X}^{i,j})$, direct exponentiation can lead to numerical overflow. To address this, we compute $\boldsymbol{l}^{i,j}$ using the following stabilized formulation:

$$\boldsymbol{l}^{i,j} = \log\sum_k e^{\boldsymbol{X}^{i,j}_{:,k}} = \boldsymbol{m}^{i,j} + \log\sum_k e^{\boldsymbol{X}^{i,j}_{:,k} - \boldsymbol{m}^{i,j}}, \tag{5}$$

where $\boldsymbol{m}^{i,j} = \max_k \boldsymbol{X}^{i,j}_{:,k}$ is a vector, with each element representing the maximum value of the corresponding row in $\boldsymbol{X}^{i,j}$. This vector acts as a normalization factor, ensuring that the values inside the exponential function remain numerically stable.

This tile-wise approach significantly reduces the memory requirement by allowing each GPU to compute and store only a subset of the similarity matrix at any given time, rather than the entire $b \times b$ matrix. Additionally, this method facilitates scaling to larger batch sizes by enabling parallel computation of the tiles on multiple GPUs or across different nodes in a distributed system.

**Tile-Wise Backward.** According to the chain rule, the gradients *w.r.t.* $\boldsymbol{I}_i$ and $\boldsymbol{T}_j$ are

$$\frac{\partial\mathcal{L}_I}{\partial\boldsymbol{I}_i} = \sum_j \frac{\partial\mathcal{L}_I}{\partial x_{i,j}} \cdot \frac{\partial x_{i,j}}{\partial\boldsymbol{I}_i}, \quad \frac{\partial\mathcal{L}_I}{\partial\boldsymbol{T}_j} = \sum_i \frac{\partial\mathcal{L}_I}{\partial x_{i,j}} \cdot \frac{\partial x_{i,j}}{\partial\boldsymbol{T}_j}. \tag{6}$$

Taking the gradients *w.r.t.* $\boldsymbol{I}_i$ as an example, according to Equation 2, the complete formulation is

$$\frac{\partial\mathcal{L}_I}{\partial\boldsymbol{I}_i} = -\frac{1}{b}\sum_j \left(\frac{\partial\mathcal{L}_I}{\partial x_{i,i}} \cdot \frac{\partial x_{i,i}}{\partial x_{i,j}} \cdot \frac{\partial x_{i,j}}{\partial\boldsymbol{I}_i} - \frac{\partial\mathcal{L}_I}{\partial l_i} \cdot \frac{\partial l_i}{\partial x_{i,j}} \cdot \frac{\partial x_{i,j}}{\partial\boldsymbol{I}_i}\right)$$

$$= -\frac{1}{b} \cdot \boldsymbol{T}_i + \frac{1}{b}\sum_j e^{x_{i,j} - l_i} \cdot \boldsymbol{T}_j. \tag{7}$$

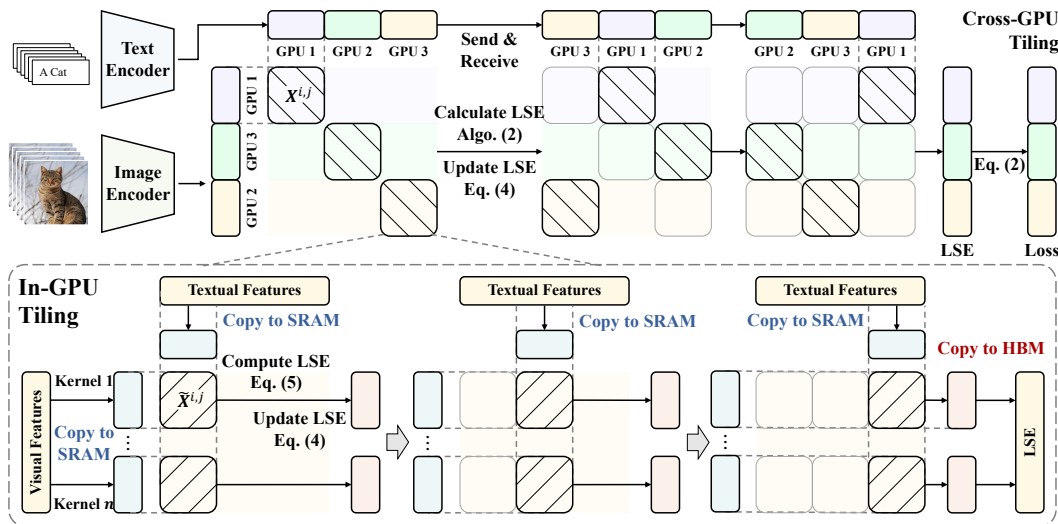

Figure 3: **Multi-level tiling strategy. Top**: for cross-GPU tiling, each GPU is assigned with multiple rows. The computation and the column-wise communication are performed asynchronously to reduce the cost. **Bottom**: for in-GPU tiling, the calculations in each GPU are further divided into tiles and the row-wise calculation is distributed to multiple CUDA cores. The accumulative operations of each row are merged into one kernel for reducing I/O times between SRAM and HBM.

From the formula, it can be seen that the second term requires the similarities $x_{i,j}$ with $\mathcal{O}(b^2)$ memory in common implementations, whether stored in the forward process or computed directly in the backward process. To tackle this, we apply the similar tile-based method as the forward process to compute the gradient. Specifically, we first store $l$, which has only $b$ elements during forward propagation, and calculate the gradient *w.r.t* $I_i$ by iterative accumulation in multiple tiles:

$$
\begin{aligned}
I_i' &\leftarrow I_i' + e^{x_{i,j}-l_i} \cdot T_j, \ j = 1, \ldots, n_c, \\
\frac{\partial \mathcal{L}_I}{\partial I_i} &= -\frac{1}{b} \cdot T_i + \frac{1}{b} I_i',
\end{aligned}
\tag{8}
$$

where $I_i'$ is a temporary variable for accumulation. The detailed algorithm is shown in Appendix.

## 3.2 Multi-Level Tiling

The scaling of batch size is usually accompanied by the scaling of the number of GPUs. In order to fully utilize the parallelism between multiple GPUs while exploiting partially serial computation on a single GPU to reduce the memory cost, we propose a multi-level tiling method that distributes the above LSE calculation to coarse-grained cross-GPU tiles and fine-grained in-GPU tiles.

**Cross-GPU Tile.** As shown in Algorithm 1, in data parallel training with $n$ GPUs, the $i$-th GPU first processes a portion of images and texts to visual features $I^i \in \mathbb{R}^{b_s \times c}$ and textual features $T^i \in \mathbb{R}^{b_s \times c}$, where $b_s = b/n$ is the batch size in one GPU. Then for the calculation of the contrastive loss, we distribute computations of different rows to different GPUs and synchronize the columns between GPUs step-by-step, considering the row-wise characteristic. Specifically, the $i$-th GPU is responsible for calculating $X^{i,:}$ and the corresponding $l^i$. For memory considerations, based on the tiling strategy described in Section 3.1 where only one tile $X^{i,j}$ is computed at a time, $X^{i,:}$ is further divided into $X^{i,j}$ for $n$ step to calculate $l^i$ following Equation 4, where the local LSE $l^{i,j}$ is calculated by in-gpu tiling as described in the next part.

Moreover, since the computation of $X^{i,j}$ while $i \neq j$ requires the textual feature $T^j$ stored in other GPUs, additional communication overhead is inevitable, especially as the number of GPUs grows. In order to reduce or even eliminate the communication overhead, we associate all GPUs with a ring topology, based on the idea of overlapping communication time and computation time overlap as much as possible. Concretely, starting with $T^i$, each GPU process sends the current textual

features to the next process and receives the textual features from the previous process using the ring topology while computing Equation 4. In this way, the communication time cost is negligible when it is greater than the computation time overhead.

---

**Algorithm 1** Forward Process of Multi-level Tile-Wise Global LSE Calculation

---

**Require:** Number of GPUs $n$, in-memory visual features $\boldsymbol{I}^i \in \mathbb{R}^{b_s \times c}$ and textual features $\boldsymbol{T}^i \in \mathbb{R}^{b_s \times c}$ for each GPU.
1: **for** $counter$ = 1 **to** $n$ **do**
2:    **Update LSE:**
3:       Each GPU computes the local LSE vector via Algorithm 2 with in-memory features.
4:       Each GPU updates the LSE vector via Equation 4.
5:    **Asynchronously Communication:**
6:       Each GPU sends the in-memory textual feature to the next GPU in the ring.
7:       Each GPU receives the textual feature from the previous GPU in the ring.
8: **end for**
9: Return the final LSE vector $\boldsymbol{l}_i$ for each GPU .

---

**In-GPU Tile.** With the cross-GPU tiling technique, the memory complexity becomes $\mathcal{O}(b_s^2)$ for directly storing $\boldsymbol{X}^{i,j}$ where $b_s = b/n$. Since the number of GPU $n$ is somehow limited, we further introduce in-GPU tiling to reduce the $\mathcal{O}(b_s^2)$ memory cost to $\mathcal{O}(b_s)$ for enabling further batch size scaling. Specifically, we first split $\tilde{\boldsymbol{X}} = \boldsymbol{X}^{i,j}$ into tiles:

$$\tilde{\boldsymbol{X}} = [\tilde{\boldsymbol{X}}^{i,j}], \ i = 1, \ldots, \tilde{n}_r, \ j = 1, \ldots, \tilde{n}_c, \tag{9}$$

where $\tilde{n}_r = \lceil b/t_r \rceil$ and $\tilde{n}_c = \lceil b/t_c \rceil$ and $t_r$ and $t_c$ is the row-wise and column-wise size of a tile. For implementation, we distribute rows to multiple CUDA cores to make full use of the parallel computing power of the GPU, and serial process the row-wise tiles in each kernel by applying Equation 5 and Equation 4 to $\tilde{\boldsymbol{X}}^{i,j}$, as shown in Algorithm 2.

The iterative computation requires multiple memory access for variable $\boldsymbol{l}^i$. To avoid expensive I/O from HBM to SRAM, we fuse the row-wise iterative calculation into one kernel. Specifically, $\boldsymbol{l}^i$ and $\tilde{\boldsymbol{X}}^{i,j}$ are allocated in SRAM. In this way, the image features are loaded to SRAM only once at beginning, and $\tilde{\boldsymbol{l}}^i$ is written to HBM only once in the end, as shown in Figure 3.

---

**Algorithm 2** Forward Process of Tile-Wise Local LSE Calculation

---

**Require:** Visual features: $\tilde{\boldsymbol{I}} \in \mathbb{R}^{b_s \times c}$ and textual features: $\tilde{\boldsymbol{T}} \in \mathbb{R}^{b_s \times c}$, the row-wise and column-wise size of a tile: $t_r$ and $t_c$.
1: Divide $\tilde{\boldsymbol{I}}$ into $\tilde{\boldsymbol{I}}^i$, where $i = 1, 2, \ldots, \tilde{n}_r$.
2: Divide $\tilde{\boldsymbol{T}}$ into $\tilde{\boldsymbol{T}}^j$, where $j = 1, 2, \ldots, \tilde{n}_c$.
3: **parallel for** each $\tilde{\boldsymbol{I}}^i$ **do**
4:    Load $\tilde{\boldsymbol{I}}^i$ from HBM to on-chip SRAM.
5:    Initialize $\tilde{\boldsymbol{l}}^i = \boldsymbol{0} \in \mathbb{R}^{t_r}$.
6:    **for** $j$ = 1 **to** $\tilde{n}_r$ **do**
7:       Load $\tilde{\boldsymbol{T}}_j$ from HBM to on-chip SRAM.
8:       On chip, compute $\tilde{\boldsymbol{X}}^{i,j} = \tilde{\boldsymbol{I}}^i \cdot \tilde{\boldsymbol{T}}^{j\prime} \in \mathbb{R}^{t_r \times t_c}$.
9:       On chip, calculate tile LSE $\tilde{\boldsymbol{l}}^{i,j}$ based on Equation 5:
10:          $\tilde{\boldsymbol{l}}^{i,j} = \tilde{\boldsymbol{m}}^{i,j} + \text{LSE}(\tilde{\boldsymbol{X}}^{i,j} - \tilde{\boldsymbol{m}}^{i,j})$, where $\tilde{\boldsymbol{m}}^{i,j} = \text{rowmax}(\tilde{\boldsymbol{X}}^{i,j})$.
11:       On chip, update LSE $\tilde{\boldsymbol{l}}^i$ based on Equation 4:
12:          $\tilde{\boldsymbol{l}}^i \leftarrow \tilde{\boldsymbol{l}}^i + \log(1 + \exp(\tilde{\boldsymbol{l}}^{i,j} - \tilde{\boldsymbol{l}}^i))$.
13:    **end for**
14:    Write $\tilde{\boldsymbol{l}}^i$ to HBM.
15: **end parallel for**
16: Return $\tilde{\boldsymbol{l}}$.

---

| Model | Loss (Peak) Memory Cost (GB) | | | | |
|---|---|---|---|---|---|
| | 32k | 64k | 128k | 256k | 1024k |
| *8×A800 (≈ 8 × 80GB)* | | | | | |
| CLIP | 16.67 (46.40) | 66.11 (77.94) | ✗ | ✗ | ✗ |
| OpenCLIP | 2.27 (43.97) | 8.63 (46.38) | 33.64 (51.23) | ✗ | ✗ |
| Inf-CL | 0.18 (44.20) | 0.36 (46.63) | 0.72 (51.46) | 1.45 (61.13) | ✗ |
| Inf-CL* | 0.18 (42.40) | 0.36 (42.49) | 0.72 (42.69) | 1.45 (43.07) | 6.53 (45.40) |
| *32×A800 (≈ 32×80GB)* | | | | | |
| CLIP | 16.66 (42.85) | 66.11 (75.52) | ✗ | ✗ | ✗ |
| OpenCLIP | 0.71 (42.46) | 2.45 (43.06) | 8.98 (44.26) | 34.35 (46.71) | ✗ |
| Inf-CL | 0.05 (42.48) | 0.09 (43.08) | 0.18 (44.30) | 0.35 (46.71) | 1.44 (61.20) |

Table 1: **Training Memory Cost Across Different Hardware and Batch Sizes**. Experiments utilize Data Parallelism with Automatic Mixed Precision for efficient distributed training. The baselines include the *Vanilla loss* (CLIP) and *Local loss* (OpenCLIP). To minimize memory consumption, Gradient Cache is adopted, with an accumulation batch size of 128. * indicates the use of the data offload strategy, which reduces memory usage by transferring only a small data batch from CPU to GPU during each accumulation step. ✗ denotes cases where the baseline exceeds the hardware memory limit for a given batch size, making training infeasible. Memory cost is evaluated using the ViT-L/14 architecture and the AdamW optimizer.

## 4 EXPERIMENTS

### 4.1 EXPERIMENTAL SETTINGS

**Dataset and Data Processing.** We assess the effectiveness of our **Inf-CL** on Laion400M dataset (Schuhmann et al., 2021) where we used 280M (out of 400M) samples for training due to the unavailability of images in the remaining samples. Images undergo preprocessing using RandomResizedCrop with a crop ratio of $[0.75, 1.33]$ and a scale of $[0.08, 1.0]$.

**Training Hyperparameters.** A modified AdaFactor optimizer (Shazeer & Stern, 2018) is employed for training, following the settings of ViT-g (Zhai et al., 2022a). The optimizer is configured with a learning rate of $1 \times 10^{-3}$, weight decay of $1 \times 10^{-4}$, and coefficients $\beta_1 = 0.9$ and $\beta_2 = 0.95$ (Zhai et al., 2023). Training spans 8 epochs, using a cosine learning rate schedule with a linear warm-up during the first 0.5 epoch.

**Implementation Details.** For distributed training, we employ Data Parallelism (Li et al., 2020) with Automatic Mixed Precision (float16)(Micikevicius et al., 2017). To support larger batch sizes, we adopt Gradient Cache (Gao et al., 2021) which decouples contrastive loss computation from the model's forward and backward passes. Consequently, the peak memory cost per iteration, $M_{peak}$, is calculated as:

$$M_{peak} \approx M_{data} + \max(M_{loss}, M_{backbone}), \tag{10}$$

where $M_{data}$ is the memory for data, $M_{loss}$ is for loss computation, and $M_{backbone}$ is for the model's forward and backward operations.

**Baselines.** We compare our method against two baselines: the *vanilla loss* from CLIP and the *local loss* from OpenCLIP/DisCo-CLIP. The *vanilla loss* computes a $b \times b$ similarity matrix by gathering both row and column features from all GPUs, while the *local loss* requires only column features to calculate a $b/n \times b$ similarity matrix, where $b$ and $n$ are the batch size and the number of GPUs.

### 4.2 COST ANALYSIS

Our method, as detailed in Section 3.2, divides the calculation of contrastive loss into tiles and distributes them across different GPUs and GPU kernels. To rigorously assess its memory efficiency, we compare our approach with previous methods like CLIP and OpenCLIP by evaluating "Memory

| Budget | Maximum Batch Size (Loss Memory Cost) | | | Improvement |
|---|---|---|---|---|
| | CLIP | OpenCLIP | Inf-CL | (Ours / Sota) |
| *ViT-B/16* | | | | |
| 8×A800 | 68k (74.39 GB) | 172k (59.95 GB) | **800k** (3.01 GB) | 4.65 (800k/172k) |
| 32×A800 | 68k (74.39 GB) | 360k (66.29 GB) | **3456k** (3.27 GB) | 9.60 (3456k/360k) |
| *ViT-L/14* | | | | |
| 8×A800 | 64k (66.11 GB) | 152k (47.23 GB) | **448k** (2.52 GB) | 2.94 (448k/152k) |
| 32×A800 | 64k (66.11 GB) | 352k (64.13 GB) | **2048k** (2.89 GB) | 5.82 (2048k/256k) |
| *ViT-L/14* w/ data offload | | | | |
| 8×A800 | 64k (66.11 GB) | 184k (69.10 GB) | **4096k** (26.12 GB) | 22.26 (4096k/184k) |
| 32×A800 | 64k (66.11 GB) | 368k (64.13 GB) | **12288k** (19.59 GB) | 33.39 (12288k/368k) |

Table 2: **Maximum batch size** for model training using different hardware and contrastive loss methods. The training setting of this experiment is aligned with Table 1.

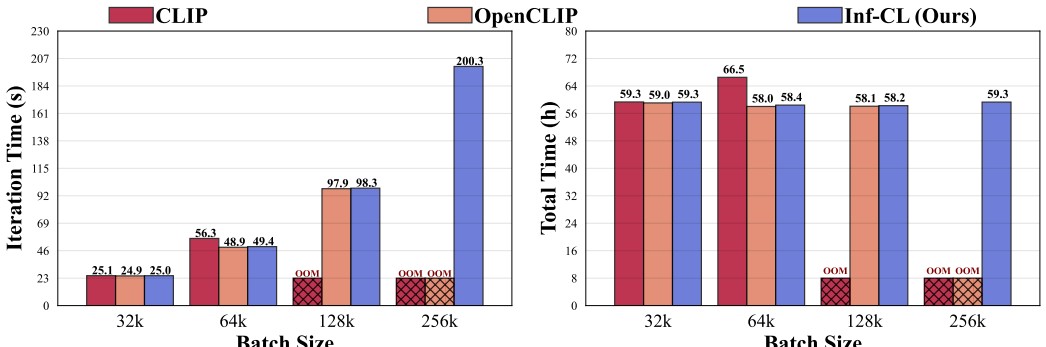

Figure 4: **Training Speed of ViT-L/14 CLIP on 8×A800 for Varying Batch Sizes.** The left figure shows the time per iteration step, while the right displays the time per epoch. Loss calculation contributes minimally to the total iteration time, making Inf-CL's iteration time comparable to previous methods. Furthermore, the iteration time of **Inf-CL** scales linearly with batch size, leading to a stable training duration of approximately 59 hours per epoch.

Consumption","Max Supported Batch Size" and "Speed" across various model architectures and hardware settings. The effective memory cost is determined by peak memory (Equation 10), which is the maximum memory needed during an iteration.

**Memory Consumption**. To illustrate the memory efficiency of Inf-CL, we compared it to previous methods using the same batch size. Table 1 shows that for loss calculation, Inf-CL requires significantly less memory than its predecessors. Specifically, with a batch size of 128k on 8×A800, Inf-CL only consumes 0.72 GB, whereas OpenCLIP requires 33.64 GB. However, while the memory cost of loss calculation with Inf-CL is minimal, peak memory usage still increases rapidly with batch size due to growing data memory, as discussed in "Max Supported Batch Size." By integrating Inf-CL with "data offload", we can mitigate this memory increase, enabling us to train a ViT-L/14 model with a batch size of 1024k on 8×A800.

**Maximum Batch Size.** We compare the maximum batch size of Inf-CL with those of previous approaches under various model architectures (ViT-B/16 or ViT-L/14) and training budgets (8×A800 or 32×A800). As shown in Table 2. Inf-CL significantly outperforms previous SOTA methods, achieving improvements of 4.65× for ViT-B/16 on 8×A800, which is further increased to 9.60× when using 32×A800. Notably, as we scale up the model size, the improvements decrease; for instance, from 4.65 to 2.94 when changing from ViT-B/16 to ViT-L/14. To understand this trend, we analyze peak memory usage. Since Inf-CL has negligible memory requirements, peak memory is primarily driven by $M_{backbone} + M_{data}$. $M_{backbone}$ is constant, meaning the rapid growth in peak

| Method (Batch Size) | ImageNet | | | | MSCOCO R@1 | |
| --- | --- | --- | --- | --- | --- | --- |
| | Validation | v2 | ObjectNet | OOD | I→T | T→I |
| Vanilla (64K) | 74.74 | **65.30** | 46.31 | 66.13 | 25.71 | 44.31 |
| OpenCLIP (64K) | 74.86 | 65.22 | 46.29 | 66.75 | 25.98 | 44.02 |
| Inf-CL (64K) | 74.93 | 65.27 | 46.13 | 66.77 | **26.01** | 43.95 |
| Inf-CL (256K) | **75.12** | 65.12 | **46.44** | **67.15** | 25.90 | **44.61** |
| Inf-CL (1024K) | 73.58 | 63.87 | 44.55 | 64.60 | 24.53 | 41.58 |

Table 3: **Performance Verification.** The training strategies is consistent with Table 2. We choose ViT-B/16 as the model architecture and adopt LiT strategy like Table 4. We evaluate zero-shot top-1 classification accuracy on several data sets, e.g., ImageNet-Validation Deng et al. (2009), ImageNet-v2 (Recht et al., 2019), ObjectNet (Barbu et al., 2019) and ImageNet-OOD (Hendrycks et al., 2021). We also evaluate zero-shot image-text top-1 retrieval accuracy on MSCOCO (Chen et al., 2015).

| Cross-GPU | In-GPU | Data | Loss | | Backbone | Peak | ImageNet |
| --- | --- | --- | --- | --- | --- | --- | --- |
| | | Memory | Complexity | Memory | Memory | Memory | |
| (Vanilla) | | 1.96 | $\mathcal{O}(b^2)$ | 66.21 | 8.26 | 69.24 | 74.82 |
| (OpenCLIP) | | 1.96 | $\mathcal{O}(b^2/n)$ | 16.96 | 8.26 | 20.79 | 74.86 |
| ✔ | | 1.96 | $\mathcal{O}(b^2/n^2)$ | 4.81 | 8.26 | 12.30 | 74.78 |
| ✔ | ✔ | 1.96 | $\mathcal{O}(b/n^2)$ | 0.81 | 8.26 | 12.30 | 74.93 |

Table 4: **Ablation Study of Multi-level Tiling Strategy.** The training strategies is consistent with Table 2, using the ViT-B/16 architecture. To reduce memory consumption and expedite experimentation, we freeze the image encoder and load pretrained weights as done in LiT. The global batch size is fixed at 64k with an accumulation batch size of 256 per GPU. These experiments are conducted on 4×A800 (80G) GPUs. "Complexity" denotes the space complexity of loss calculation. $b$ denotes batch size, while $n$ denotes the number of GPUs.

memory is mainly due to increased $M_{data}$. Since ViT-L/14 has a larger $M_{backbone}$, the remaining memory can accommodate only a smaller batch size for $M_{data}$. To address this issue, we implement "data offload", which allows us to load only a small batch of data onto the GPU for each accumulation step, effectively stabilizing the data memory usage. Therefore, by combining data offload with our Inf-CL, we can scale the batch size to over 10M on 32×A800.

**Training Speed.** We compare the training speed of our Inf-CL with previous methods. As shown in Figure 4, using Inf-CL to train ViT-L/14 on 8×A800 has almost the same speed as previous methods. Even when increasing batch size beyond the limits of previous methods, Inf-CL maintains a linear increase in iteration time, with one epoch consistently taking about 59 hours. Combining training speed results with memory cost results demonstrates that our Inf-CL has superior memory efficiency, while only introducing a little additional time cost (extra analysis in Appendix A.2).

### 4.3 PERFORMANCE ANALYSIS

In this section, we investigate whether introducing Inf-CL negatively affects CLIP performance and whether increasing batch size with Inf-CL enhances performance. Due to the limit of GPU resources, we utilize the ViT-B/16 with Bert-Base (Devlin, 2018). We follow the training strategy of LiT (Zhai et al., 2022b) to freeze the visual backbone and use the pre-trained weights instead.

**Performance Verification**. We evaluate CLIP models trained with different loss implementations, with the results presented in Table 3. As shown, under the same batch size, our Inf-CL performs similarly to previous methods, with performance differences falling within the error margin, confirming that our design incurs no precision loss in the loss calculations. Furthermore, the results indicate that increasing the batch size within a certain range yields performance enhancements, thereby underscoring the significance of our method for helping scale the batch size. However, under our

experimental conditions, we currently observe that an excessively large batch size—previously un-examined in the literatures—results in suboptimal performance. This may be attributed to factors such as unoptimized hyperparameters, inadequate training iterations, or constraints related to data size (for a comprehensive analysis, see Appendix A.3). Since our work mainly focus on how to enable large batch size training, these factors warrant further investigation in future work.

**Ablation Study**. We ablate multi-level tiling in Table 4 and show that our designs incur no precision loss in loss calculations. This allows arbitrary combinations to achieve nearly the same zero-shot classification accuracy (about 74.8% on ImageNet for 64k batch size), while significantly reducing memory costs. According to the Equation 10, their $M_{peak}$ is decided by $M_{backbone} + M_{data}$ rather than $M_{loss} + M_{data}$ as in prior methods. For complexity analysis, Cross-GPU tiling is $\mathcal{O}(b^2/n^2)$, resulting in a memory cost that is $1/n$ of OpenCLIP ($16.96/4.81 \approx 4$ in Table 4). Based on it, introducing In-GPU tiling can further reduce memory cost and make the growth of memory cost linear, i.e., $\mathcal{O}(b^2/n^2) \rightarrow \mathcal{O}(b/n^2)$.

## 5 RELATED WORK

**Contrastive Learning:** The core idea of contrastive learning is to learn better representations by distinguishing between positive and negative pairs of samples (van den Oord et al., 2018; Chen et al., 2020b). This approach demonstrates strong effectiveness across diverse tasks, as the nature of the paired samples varies depending on the specific application. In image foundation models, such as SimCLR (Chen et al., 2020a) and MoCo (He et al., 2020), positive pairs are created by augmenting the same image in different ways. For cross-modal retrieval, as exemplified by CLIP (Radford et al., 2021) and ALIGN (Jia et al., 2021), the positive pairs consist of aligned image and text samples. Similarly, for dense text retrieval (Karpukhin et al., 2020; Wang et al., 2022; Zhang et al., 2022), the positive pairs are composed of query and document pairs. Several works improve contrastive learning performance by enhancing dataset quality, modifying the loss function, or refining negative sample selection (Vasu et al., 2024; Zhai et al., 2023; Zhang et al., 2023). Moreover, several studies, both empirical and theoretical, have demonstrated from various perspectives that larger batch sizes contribute to learning better representations (Saunshi et al., 2019; Chen et al., 2022). Due to the quadratic growth of memory usage with batch size in classical contrastive loss, most existing studies have stopped scaling their batch sizes to 128k, even when leveraging hundreds of GPUs (Radford et al., 2021; Jia et al., 2021; Yang et al., 2022).

**Memory-efficient Training:** As deep learning models continue to grow in size and complexity, the demand for computational resources, particularly GPU memory, has increased significantly. Techniques such as Gradient Checkpointing (Sohoni et al., 2022) recompute activations during back-propagation to save memory at the expense of additional computation. Flash Attention (Dao et al., 2022) reduces memory overhead by computing attention in blocks without storing large interme-diate states. Ring Attention (Liu et al., 2023) distributes long sequence activations across multiple devices, overlapping computation and communication to train sequences far longer than previous methods. For contrastive learning, GradCache (Gao et al., 2021) and BASIC (Pham et al., 2021) in-troduce a gradient caching technique that decouples backpropagation between contrastive loss and the encoder, which reduces memory usage in the model by accumulating gradients per mini-batch. OpenCLIP (Ilharco et al., 2021) and DisCo-CLIP (Chen et al., 2023) reducing memory consumption by distributing the computation of contrastive loss across multiple GPUs.

## 6 CONCLUSION

This paper addresses the GPU memory bottleneck in scaling batch sizes for contrastive loss. To overcome the quadratic memory consumption resulting from the full instantiation of the similarity matrix, we proposed a tile-based computation strategy that partitions the calculation into smaller blocks, thus avoiding full matrix materialization. Furthermore, we introduced a multi-level tiling strategy that leverages ring-based communication and fused kernels to optimize synchronization and minimize I/O overhead. Our experiments demonstrated that our method scales contrastive loss batch sizes to unprecedented levels without compromising accuracy or training speed. This approach marks a significant advancement in large-scale contrastive learning, shedding light on further devel-opments in areas such as self-supervised learning and dense text retrieval.

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

## A    APPENDIX

### A.1    BACKWARD PROCESS

---

**Algorithm 3** Backward Process of Multi-level Tile-Wise Global LSE Calculation

---

**Require:** Number of GPUs $n$, saved intermediate variables from the forward pass: in-memory
visual features $\boldsymbol{I}^i \in \mathbb{R}^{b_s \times c}$ and textual features $\boldsymbol{T}^i \in \mathbb{R}^{b_s \times c}$ for each GPU, global LSE vectors
$\boldsymbol{l}^i \in \mathbb{R}^{b_s}$.
1: Initialize vector: $\boldsymbol{dI}^i = \boldsymbol{0} \in \mathbb{R}^{b_s \times c}$, $\boldsymbol{dT}_{\text{cache}} = \boldsymbol{0} \in \mathbb{R}^{b_s \times c}$ on each GPU$_i$.
2: **for** $j = 1$ **to** $n$ **do**
3:     **Asynchronously Text Feature Communication:**
4:     Each GPU sends in-memory textual feature to the next GPU and receive the textual
    feature from the previous GPU in the ring.
5:     **Backward Calculation:**
6:     Index of current text feature tile for each GPU: $k = (i + j - 1) \mod n$
7:     Call Algorithm 4 with $(\boldsymbol{I}^i, \boldsymbol{T}^k, \boldsymbol{l}^i)$ , obtaining gradients $\boldsymbol{dI}^i_{\text{temp}}$ and $\boldsymbol{dT}^k_{\text{temp}}$.
8:     Update gradients $\boldsymbol{dI}^i \mathrel{+}= \boldsymbol{dI}^i_{\text{temp}}$.
9:     Update gradients $\boldsymbol{dT}_{\text{cache}} \mathrel{+}= \boldsymbol{dT}^k_{\text{temp}}$.
10:    **Asynchronously Gradient Communication:**
11:    Each GPU sends in-memory $\boldsymbol{dT}_{\text{cache}}$ to the next GPU in the ring.
12:    Each GPU receive the gradient feature from the previous GPU and write to $\boldsymbol{dT}_{\text{cache}}$.
13: **end for**
14: $\boldsymbol{dT}^i = \boldsymbol{dT}_{\text{cache}}$ in each GPU.
15: Return the gradients $\boldsymbol{dI}^i$, $\boldsymbol{dT}^i$ for each GPU.

---

---

**Algorithm 4** Backward Process from of intra-GPU Tile-Wise LSE calculation

---

**Require:** Saved intermediate variables from the forward pass: visual features $\tilde{\boldsymbol{I}} \in \mathbb{R}^{b \times c}$, textual
features $\tilde{\boldsymbol{T}} \in \mathbb{R}^{b \times c}$, the local LSE vector $\tilde{\boldsymbol{l}} \in \mathbb{R}^b$.
    The row-wise and column-wise size of a tile: $t_r$ and $t_c$,
1: Divide $\tilde{\boldsymbol{I}}$ into $\tilde{\boldsymbol{I}}^i$, where $i = 1, 2, \ldots, \tilde{n}_r$.
2: Divide $\tilde{\boldsymbol{T}}$ into $\tilde{\boldsymbol{T}}^j$, where $j = 1, 2, \ldots, \tilde{n}_c$.
3: Divide $\tilde{\boldsymbol{l}}$ into $\tilde{\boldsymbol{l}}^i$, where $i = 1, 2, \ldots, \tilde{n}_r$.
4: Initialize gradients vectors: $\boldsymbol{d\tilde{I}} \in \mathbb{R}^{t_r \times c}$ and $\boldsymbol{d\tilde{T}} \in \mathbb{R}^{t_c \times c}$.
5: **for** each $\tilde{\boldsymbol{I}}^i$ **do**
6:     Load $\tilde{\boldsymbol{I}}^i$ and $\tilde{\boldsymbol{l}}^i$ from HBM to on-chip SRAM.
7:     Initialize $\boldsymbol{d\tilde{I}}^i = \boldsymbol{0} \in \mathbb{R}^{t_r \times c}$.
8:     **for** $j = 1$ **to** $[b // t_c]$ **do**
9:        Load $\tilde{\boldsymbol{T}}^j$ from HBM to on-chip SRAM.
10:      On chip, compute $\tilde{\boldsymbol{X}}^{i,j} = \tilde{\boldsymbol{I}}^i \cdot \tilde{\boldsymbol{T}}^{j'} \in \mathbb{R}^{t_r \times t_c}$.
11:      On chip, compute $\boldsymbol{d\tilde{X}}^{i,j} = \exp(\tilde{\boldsymbol{X}}^{i,j} - \tilde{\boldsymbol{l}}^i) \in \mathbb{R}^{t_r \times t_c}$.
12:      Update gradients $\boldsymbol{d\tilde{I}}^i \mathrel{+}= \boldsymbol{d\tilde{X}}^{i,j} \cdot \tilde{\boldsymbol{T}}^j$.
13:      Load $\boldsymbol{d\tilde{T}}^j$ from HBM to on-chip SRAM.
14:      $\boldsymbol{d\tilde{T}}^j \mathrel{+}= \tilde{\boldsymbol{I}}^i \cdot \boldsymbol{d\tilde{X}}^{i,j}$.
15:      Write updated $\boldsymbol{d\tilde{T}}^j$ back to HBM.
16:     **end for**
17:     Write updated $\boldsymbol{d\tilde{I}}^i$ back to HBM.
18: **end for**
19: **return** $\boldsymbol{d\tilde{I}}$(i.e. $\frac{\partial \tilde{\boldsymbol{l}}}{\partial \tilde{\boldsymbol{I}}}$), $\boldsymbol{d\tilde{T}}$($i.e. \frac{\partial \tilde{\boldsymbol{l}}}{\partial \tilde{\boldsymbol{T}}}$).

---

### A.2    ANALYSIS OF TRAINING SPEED EFFICIENCY IN INF-CL

Although **Inf-CL** might be expected to exhibit slower performance because it breaks the loss cal-
culation to small tiles and serially process these tiles, it achieves comparable speed to previous

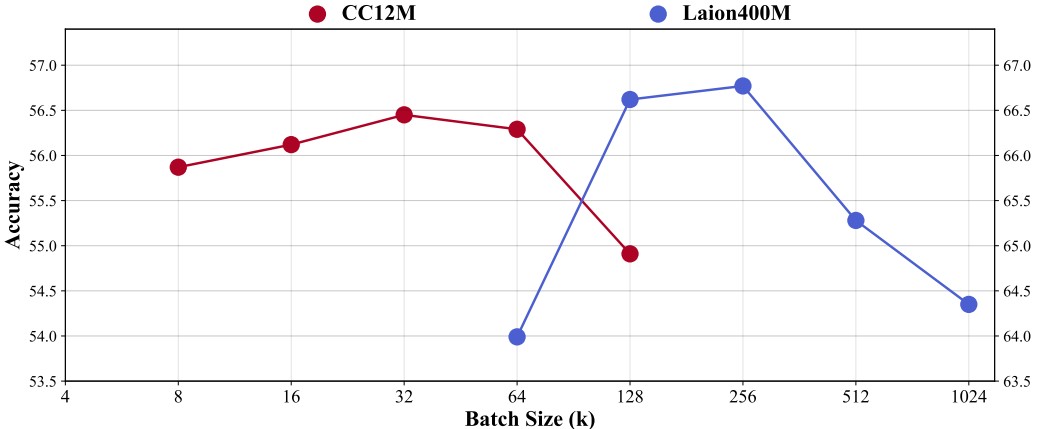

Figure 5: **Performance of ViT-B/32 across Varying Batch Sizes**. Except batch size, other experiment settings ar e consistent. ❶ Left: On CC12M, performance plateaus at a batch size of approximately 32k. ❷ Right: On Laion400M, performance saturates around a batch size of 256k.

methods, as shown in Figure 4. This is primarily due to two factors: (1) Loss calculation represents only a minor fraction of the total iteration time, especially for large models, thereby exerting minimal impact on the overall iteration time. (2) While **Inf-CL** has similar computational complexity to standard contrastive loss, its tiling approach could introduce some speed overhead due to reduced parallelism. However, **Inf-CL** fuses the operations of similarity matrix calculation and softmax, which in regular contrastive loss require two separate communications between SRAM and HBM. By merging these into a single communication, **Inf-CL** effectively reduces I/O time, mitigating the cost of serial tile computation.

### A.3 Factors influencing performance when scaling batch size

While larger batch size is theoretically expected to enhance performance Chen et al. (2022), our experimental results deviate from this expectation. To better understand this discrepancy, we analyze the factors that impact performance when scaling up batch size.

**Hyperparameters**. Although larger batch sizes provide more diverse negative samples for contrastive learning, potentially improving the embedding space, careful tuning of hyperparameters is necessary to ensure model convergence. Previous research indicates that when increasing batch size, the learning rate should be scaled proportionally to maintain a consistent parameter update norm throughout training (Goyal, 2017). Since a fixed learning rate is used across all experiments, this may have contributed to the reduced performance observed with larger batch sizes. Moreover, prior studies suggest that large batch sizes require longer training epochs to ensure sufficient parameter updates and avoid suboptimal convergence (Hoffer et al., 2017). Overall, the performance gains from larger batch sizes are contingent on the careful tuning of multiple hyperparameters beyond just learning rate and epochs, highlighting the importance of comprehensive hyperparameter optimization to fully exploit the benefits of scaling.

**Data Scale.** Increasing batch size improves the precision of gradient estimation for the representation distribution defined by the dataset Chen et al. (2022). Larger datasets more accurately capture real-world distributions, and thus, employing a larger batch size enables contrastive loss to generate more precise gradients, enhancing the model's ability to learn discriminative representations. As shown in Figure 5, our experiments on different data scales (e.g., CC12M and Laion400M) indicate that the optimal batch size increases with dataset size. Specifically, performance on CC12M saturates at a batch size of 32k, whereas Laion400M achieves saturation at a batch size of 256k.

In summary, while scaling up batch sizes is critical for enhancing contrastive learning, our findings suggest that performance does not monotonically improve with batch size increases. As seen in our previous experiments (Table 3), extremely large batch sizes (e.g., 1024k) can lead to a decline in performance, indicating that factors such as hyperparameter tuning and dataset scale are among

the many considerations that influence model effectiveness. This highlights the need for a balanced approach when increasing batch sizes, ensuring that optimal configurations are found to fully exploit the benefits of contrastive learning.

