# OpenReview forum: "Breaking the Memory Barrier: Near Infinite Batch Size Scaling for Contrastive Loss"
_ICLR.cc/2025/Conference — ICLR 2025 Conference Withdrawn Submission_

### Official Review · Reviewer_fHD1 · 2024-10-18

**Soundness:** 3
**Presentation:** 2
**Contribution:** 3
**Rating:** 5
**Confidence:** 5

**Summary:**

The authors introduce Inf-CL, a way to drastically improve the (parallel) computation of contrastive losses and to scale this computation to massive batch sizes. The core issue with the current way of computing the constrastive loss is its quadratic memory consumption that is completely materialized in-memory. In this work, the authors instead propose a block-wise tiled computation of this quadratic matrix. As a result, as long as the block fits into memory, the entire contrastive loss matrix may be computed. Block sizes can be chosen arbitrarily small. Furthermore, they propose a ring communication scheme for distributed-memory system, e.g. a multi-GPU setups. In an experimental evaluation on typical benchmark datasets and models they achieve identical predictive performance, while being able to drastically increase the batch size/size of the constrastive loss matrix.

**Strengths:**

- Clear and concise writing, logical structure of the manuscript

- Meaningful visual items to explain the algorithm's principles and experimental results

- More than solid improvements on the memory footprint for computing the contrastive loss

- No loss in accuracy of the algorithm

- Intention to release the source code for reproducibility purposes

**Weaknesses:**

- It would be meaningful to have the first figure after the abstract. The authors are kindly requested to rearrange both textual elements

- The manuscript's claim of "near-infinite" scaling seems overly sensationalist. Could the authors possibly soften the wording?

- While not conceptually wrong, Algorithm 1 requires several inputs that are not used in pseudo code. It would seem meaningful to explicitly show the use of the required inputs.

- Violation of the conference template, e.g.
    * Caption of tables are above
    * Table content should not be an image but selectable text

- Computing a (symmetric) distance matrix (here: a loss, but algorithmically identical) in parallel on distributed-memory system is a well-studied problem. In principle the communication scheme boils down to a merry-go-round principle, even applied to other matrix computation problems. It would be meaningful to add at least the following works and credit them:
    * Bischof, C. H. (1989). Computing the singular value decomposition on a distributed system of vector processors. Parallel Computing, 11(2), 171-186.
    * Man, D., Uda, K., Ueyama, H., Ito, Y., & Nakano, K. (2010, November). Implementations of parallel computation of Euclidean distance map in multicore processors and GPUs. In 2010 First International Conference on Networking and Computing (pp. 120-127). IEEE.
    * Al-Neama, M. W., Reda, N. M., & Ghaleb, F. F. (2014). An improved distance matrix computation algorithm for multicore clusters. BioMed research international, 2014(1), 406178.
    * Angeletti, M., Bonny, J. M., & Koko, J. (2019). Parallel euclidean distance matrix computation on big datasets.
    * Götz, M., Debus, C., Coquelin, D., Krajsek, K., Comito, C., Knechtges, P., ... & Streit, A. (2020, December). HeAT–a distributed and GPU-accelerated tensor framework for data analytics. In 2020 IEEE International Conference on Big Data (Big Data) (pp. 276-287). IEEE.

- In the same vein, please correctly credit people for proposing communication schemes like ring-allreduce or similar. In particular:
    * Rabenseifner, R. (2004). Optimization of collective reduction operations. In Computational Science-ICCS 2004: 4th International Conference, Kraków, Poland, June 6-9, 2004, Proceedings, Part I 4 (pp. 1-9). Springer Berlin Heidelberg.

**Questions:**

- While the memory footprint reductions are impressive, how meaningful are large to huge batch sizes for models like (Open-)CLIP? Would a large batch size practically not often lead to worse out-of-sample performance?

- The authors correctly point out the symmetry of the contrastive loss, but do not meaningfully exploit this property. If the loss is symmetric, so is the resulting distance matrix. As a result the upper, respectively lower triangle of the matrix are identical to one another. Hence, the computation scheme could be further improved by only calculating on of the two and replicate the other. Have the authors considered this? If so, why is the presented algorithm not implemented as such?

---

### Official Review · Reviewer_fN3Z · 2024-11-02

**Soundness:** 3
**Presentation:** 3
**Contribution:** 2
**Rating:** 5
**Confidence:** 3

**Summary:**

Contrastive language-image learning can require O(b²) memory for a batch size of b, which includes both image and text features. This quadratic memory requirement may pose challenges for large batches. The paper addresses this limitation through a tile-based computation strategy that partitions the contrastive loss calculation into smaller blocks. Additionally, a hierarchical blocking structure has been developed for multi-GPU systems. Experiments have demonstrated that this proposed blocking approach significantly reduces memory requirements while maintaining comparable accuracy and training time.

**Strengths:**

1.	The paper clearly demonstrates that tiling effectively reduces the memory requirements for contrastive language-image learning. When large batches are desired, the proposed approach offers a practical and scalable solution to this memory challenge.
2.	Inf-CL explores various approaches to prevent numerical overflow.
3.	The multi-level algorithm uses a ring communication algorithm to share features among GPUs. This approach enables overlapping the communication with computations.

**Weaknesses:**

1. The paper does not clearly establish the necessity for large batches. The experimental results presented indicate that larger batch sizes do not lead to improvements in either accuracy or runtime. For instance, Figure 5 in the appendix shows that Laion400M reaches saturation at a batch size of 256k. This raises the question of why experiments were conducted with a batch size of 12,288k, as shown in Table 2. Additionally, Cherti et al. demonstrated (in Table 12 of their paper) that performance variations due to changes in batch size are minimal, typically in the range of 0.2% to 0.5% across different settings. Consequently, it remains unclear how the proposed algorithm would be beneficial for contrastive learning.
Ref: Cherti, Mehdi, Romain Beaumont, Ross Wightman, Mitchell Wortsman, Gabriel Ilharco, Cade Gordon, Christoph Schuhmann, Ludwig Schmidt, and Jenia Jitsev. "Reproducible scaling laws for contrastive language-image learning." In Proceedings of the IEEE/CVF Conference on Computer Vision and Pattern Recognition, pp. 2818-2829. 2023.

2. All experiments in the paper were conducted using only one dataset (Laion404M), making it unclear whether the results would hold for larger datasets referenced in the OpenCLIP benchmark.

3. The paper primarily focuses on two models, ViT-L/14 and ViT-B/16, while the OpenCLIP benchmark indicates that other models yield better results. It might be useful to clarify why the reason behind selecting ViT-L/14 and ViT-B/16 for this paper.

4. The title of the paper may be misleading, as it does not address general contrastive loss but focuses on contrastive language-image learning. It might be helpful for clarity if the title reflects the specific contributions of the paper.

**Questions:**

1.	Is there a clear benefit of using large batches in terms of accuracy and training time?
2.	Figure 5 in the appendix indicates that Laion400M reaches saturation at a batch size of 256k. What is the justification for using a batch size greater than 256k for this dataset?
3.	Can we expect to see similar performance with other models and datasets? What level of performance should we anticipate when utilizing thousands of GPUs?
4.	Although the authors mention that larger batches provide a diverse set of negative samples,  negative sampling was not used in the experiments? Does negative sampling have an impact on accuracy?

---

### Official Review · Reviewer_RxfS · 2024-11-05

**Soundness:** 3
**Presentation:** 2
**Contribution:** 2
**Rating:** 3
**Confidence:** 4

**Summary:**

The authors propose a technique Inf-CL for scaling the computations of the Contrastive Loss (CL) for Image-Text tasks. The vanilla CL relies on computing a similarity matrix between Image and Text pairs which has quadratic memory cost with respect to batch size.

With large batch sizes, that is necessary for improving the performance of Image-text CL, such similarity matrix computation becomes a bottleneck. As a result, there have been recent attempts to distribute the construction and computation of similarity matrix and the resulting CL task.

The proposed technique linearizes the memory cost of similarity matrix by employing  several optimizations such as multi-level tiling and ring all-reduce for coordinating the tasks across distributed GPU devices. In addition, the computation cost scales linearly with batch size with fine-grained tiling with smaller tiles.

**Strengths:**

1. The work tackles an important problem for Contrastive Loss (CL), i.e., how to efficiently store and compute similarly matrix that is intrinsically quadratic in batch size. The resulting impact is on how well the CL for image-text task could scale with large batch sizes for large CLIP-ViT models without sacrificing any accuracy. The empirical results demonstrate that proposed technique reduced memory cost by 78x for batch size 256k.

2. The paper is mostly clearly written with some scope of further improvement.

3. The experiments are extensive and informative supporting the claimed contributions.

**Weaknesses:**

1. The techniques used multi-level tiling and ring all-reduce are well known optimization tricks in hardware and system communities. the authors employ it in novel setup of CL with Image-Text pairs. In principle, one can argue that any task (not necessarily CL) that requires construction of a symmetric similarity matrix having quadratic memory cost would employ such strategies at core system level - be it distributing across devices or across internal cores/threads. For instance, in ViT models, $QK^T$ is a similarity matrix between queries and keys that is quadratic is number of tokens. Moreover, softmax() is also applied row-wise similar to loss computation in CL task (Equation 1 in paper). As a consequence, such optimization strategies would be standard treatment in parallelizing such tasks.

2. Tiling is a prevalent loop transformation technique that enhances both parallelism and data locality. High-performance implementations often utilize multiple levels of tiling to take full advantage of the parallelism hierarchy and optimize cache and register efficiency. Please refer to this missing reference "Multi-level tiling: M for the price of one" https://dl.acm.org/doi/abs/10.1145/1362622.1362691 that talk about multi-level tiling. As a result, I am not very convinced about the claims that Multi-level tiling is a novel contribution in the manuscript where authors state: "We propose a multi-level tiling strategy for a distributed training system, which reasonably leverages parallelism to achieve a balance between memory and computational efficiency." (Line 114-115) and Section 3.2 dedicated to Multi-Level tiling.  An extensive list of references from systems community is missing that talks about loop tiling and multi-level tiling.

**Questions:**

Q1. With respect to Lines 114-115, the authors must elaborate on what specific aspects of their multi-level tiling implementation (which exists in literature) they consider novel or uniquely suited to contrastive learning tasks ? IN current presentation, it seems the authors propose multi-level tiling.

Q2. The authors must discuss how their approach relates to or builds upon key works in loop tiling and multi-level tiling from the systems community https://scholar.google.com/scholar?hl=en&as_sdt=0%2C3&q=Multi-level+tiling&btnG= and https://scholar.google.com/scholar?hl=en&as_sdt=0%2C3&q=loop+tiling&btnG=

Q3. Is the proposed approach unique to CL or generally applicable to any task that requires construction of similar matrix with quadratic cost? In that spirit, the optimization techniques like multi-level tiling and ring all-reduce should already be used for distributing computations as core system level optimization.  Please discuss potential applications of the proposed method to other tasks involving quadratic-cost similarity matrices, and to explain any unique challenges or adaptations required for contrastive learning.

Q4. What is impact of asynchronous communication on the performance ?

Q5. Figure 2: It is challenging to observe the color schemes in tiles and with printed version, the colors wash out. The authors are encouraged to use deeper colors which are print friendly.

---

### Official Review · Reviewer_K1W1 · 2024-11-08

**Soundness:** 3
**Presentation:** 3
**Contribution:** 2
**Rating:** 5
**Confidence:** 3

**Summary:**

The authors propose a tile-based computation strategy to avoid full instantiation of the similarity matrix, which is crucial for efficiently handling large batch sizes in contrastive learning without hitting memory limits. This approach seems impactful and demonstrate that Inf-CL achieves significant reduction in memory use compared to baseline.

**Strengths:**

The paper is well-written, presenting its ideas clearly and systematically. It effectively balances theoretical insights with practical experimentation, demonstrating a strong command of both. The inclusion of detailed figures and tables enhances readability and helps illustrate the method’s advantages.

**Weaknesses:**

W1. The method's core motivation appears to stem directly from the tiling strategy detailed in the FlashAttention paper, raising questions about its originality. Furthermore, while the implementation borrows parameters from SigLip, there is no in-depth comparison or critical evaluation of SigLip, leaving its relative contributions unclear. Several other key comparisons like with DisCo-Clip is missing in the experimental section.

W2. The results show diminishing returns with increasing batch sizes, eventually leading to performance decline, as also observed in the SigLip paper. This raises doubts about the claim of "infinite batch scaling," which seems overstated. Additionally, the reasons provided in the appendix for this decline e.g. unoptimized hyperparameters and limited training, does not sufficiently supported by experimental evidence.

W3. While the paper emphasizes memory reduction through tiling strategies, it fails to detail the exact computational overhead introduced by tiling. Without reporting the specific time costs associated with tiling, beyond overall training time, it is difficult to assess the true benefits of the proposed method.

**Questions:**

1.	Can you provide statistical measures such as the mean and standard deviation for the results.
2.	Why does the paper focus only on image-text contrastive learning task?

---

### Note · Authors · 2024-11-15

I have read and agree with the venue's withdrawal policy on behalf of myself and my co-authors.